# Harnessing Biomaterials for Safeguarding Chimeric Antigen Receptor T Cell Therapy: An Artful Expedition in Mitigating Adverse Effects

**DOI:** 10.3390/ph17010139

**Published:** 2024-01-22

**Authors:** Zhaozhao Chen, Yu Hu, Heng Mei

**Affiliations:** 1Institute of Hematology, Union Hospital, Tongji Medical College, Huazhong University of Science and Technology, 1277 Jiefang Avenue, Wuhan 430022, China; zhaozhaochen@hust.edu.cn; 2Hubei Clinical Medical Center of Cell Therapy for Neoplastic Disease, Wuhan 430022, China

**Keywords:** CAR-T cell therapy, CRS, ICANS, OTOT, biomaterials, safety enhancement strategies

## Abstract

Chimeric antigen receptor T cell (CAR-T) therapy has emerged as a groundbreaking approach in cancer treatment, showcasing remarkable efficacy. However, the formidable challenge lies in taming the formidable side effects associated with this innovative therapy, among which cytokine release syndrome (CRS), immune effector cell-associated neurotoxicity syndrome (ICANS) and on-target off-tumor toxicities (OTOT) are typical representatives. Championing the next frontier in cellular immunotherapy, this comprehensive review embarks on an artistic exploration of leveraging biomaterials to meticulously navigate the intricate landscape of CAR-T cell therapy. Unraveling the tapestry of potential toxicities, our discourse unveils a symphony of innovative strategies designed to elevate the safety profile of this revolutionary therapeutic approach. Through the lens of advanced medical science, we illuminate the promise of biomaterial interventions in sculpting a safer and more efficacious path for CAR-T cell therapy, transcending the boundaries of conventional treatment paradigms.

## 1. Introduction

CAR-T therapy has revolutionized the therapeutic landscape of cancer, offering a paradigm shift in the treatment of hematologic malignancies [1]. The remarkable success of CAR-T therapy in achieving durable responses, particularly in refractory and relapsed (R/R) CD19+ leukemia and lymphoma, as well as B cell maturation antigen (BCMA)+ multiple myeloma [2,3,4,5,6,7,8,9,10,11], has underscored its potential to transform the trajectory of cancer care. However, this groundbreaking approach is not without its challenges, as the clinical translation of CAR-T cells introduces formidable side effects that necessitate careful consideration and innovative solutions [12].

The adverse events post CAR-T cell infusion encompass CRS, ICANS, OTOT, secondary haemophagocytic lymphohistiocytosis or macrophage activation syndrome (sHLH/MAS), infectious susceptibility, coagulopathy, hematologic cytopenia, latent human herpesvirus 6 reactivation, etc. [12,13,14,15,16,17,18]. At the forefront of these challenges are the intricate dynamics of adverse events, prominently exemplified by CRS, ICANS and OTOT. Various strategies have been devised to mitigate these adverse effects in efforts to alleviate these toxicities [19,20,21,22]. However, achieving the maximal cytotoxic potential of CAR-T cells while concurrently minimizing toxic side effects remains a challenging endeavor.

Biomaterials have been used in various biomedical fields, encompassing a diverse array of entities, ranging from inorganic and organic nanoparticles [23] to synthetic materials [24], hydrogels [25], and 3D scaffolds [26] that exhibit notable advantages, including strong targeting capabilities, sustained release effects, enhanced bioavailability, controlled responsivity, substantial drug-loading capacity, multifunctionality, ease of manipulation and customization, and minimal toxicity and adverse reactions [27,28,29,30]. This comprehensive review embarks on an artful expedition into the integration of biomaterials as a strategic modality to mitigate CAR-T-related adverse effects, thus fortifying the therapeutic potential of CAR-T cells. The juxtaposition of the exquisite efficacy of CAR-T therapy against the challenges posed by its side effects establishes the framework for our exploration. Through a meticulous examination of the evolving landscape of biomaterial-based interventions, we seek to unravel the tapestry of potential toxicities associated with CAR-T therapy and present an in-depth analysis of innovative strategies designed to elevate its safety profile.

As we navigate the intricate landscape of CAR-T cell therapy, the multifaceted nature of these toxicities becomes apparent. Traditional therapeutic paradigms often fall short in addressing the unique challenges posed by the balance of efficacy and toxicity [31,32]. Therefore, our discourse aims to unveil a symphony of innovative strategies grounded in the principles of biomaterial science to transcend the boundaries of conventional treatment paradigms. By doing so, we envision not only a refinement of CAR-T therapeutic strategies but also a broader impact on the field of cellular immunotherapy.

The lens through which we examine these challenges is that of advanced medical science, seeking to illuminate the promise of biomaterial interventions in sculpting a safer and more efficacious path for CAR-T cell therapy. By amalgamating the artistry of biomedical innovation with the precision of scientific inquiry, we strive to catalyze a transformative shift in the perception and application of CAR-T therapy. In the subsequent sections of this review, we will delve into the mechanisms underlying CRS, ICANS and OTOT, exploring the role of biomaterials in mitigating these adverse effects and providing a nuanced perspective on their potential clinical implications.

## 2. Mechanisms of CAR-T-Related Toxicities

### 2.1. Pathophysiological Mechanism of CRS

CRS, also known as a cytokine storm, is a systemic hyperinflammatory response triggered by the infusion of CAR-T cells into the body and their interaction with target tumor cells and other immune cells [33,34]. Typically manifesting within days post-treatment, this process involves the activation and proliferation of CAR-T cells, leading to the release of a substantial amount of inflammatory cytokines, including interleukin-6 (IL-6), interleukin-1β (IL-1β), tumor necrosis factor-α (TNF-α), interleukin-2 (IL-2), interferon-γ (IFN-γ), among others [35,36]. These released cytokines go on to further stimulate immune cells, including monocytes, macrophages, T cells, B cells, dendritic cells (DCs), natural killer (NK) cells, as well as non-immune cells such as endothelial cells and tissue cells [35,37]. This intricate network of cellular interactions leads to a cascading release of a plethora of additional cytokines, fostering a milieu conducive to heightened immune activation, thereby underscoring the complex and dynamic nature of the immune response initiated by CAR-T cell therapy.

In this context, the subsequent release of an extensive array of cytokines serves as a key orchestrator in the intricate symphony of immune activation and the ensuing clinical manifestations. CRS manifests across a spectrum of clinical symptoms, with severity ranging from mild to life-threatening [38,39]. Mild CRS symptoms typically include a low to moderate fever, mild fatigue, occasional headaches, and muscle pain. As the intensity of CRS progresses to the moderate stage, patients may experience a persistent or increasing fever, low blood pressure requiring medical attention, elevated heart rate, and mild respiratory distress. In severe cases of CRS, patients may exhibit high and persistent fever, severe hypotension leading to shock, significant respiratory distress, and multi-organ dysfunction, including liver and kidney impairment [38,39]. Such severe manifestations demand immediate medical intervention, often in an intensive care setting. Based on the severity of CRS, it can be classified into grades 1 through 4, with higher grades indicating more severe manifestations [40].

The pathophysiological and biological mechanisms underlying CRS are highly complex and remain incompletely elucidated to date (Figure 1). Several studies have endeavored to explore this intricate facet, although a comprehensive understanding has yet to be achieved. Morgane and her team investigated the roles of different CAR-T cell subsets in the occurrence of CRS in an immunocompetent mouse model, and they reported that CD4+, rather than CD8+ CAR-T cells, played an important role in inducing CRS in mice [41]. The incidence and severity of CRS were significantly correlated with the high absolute values of CD4+ CAR-T cells in the blood [41]. Theodoros and coworkers revealed that CAR-T-therapy-induced CRS was mainly mediated by IL-6, IL-1, and nitric oxide (NO) produced by recipient macrophages, not only CAR-T cells in a SCID-beige murine model of CRS [33]. They further discovered that CAR-T cells activated macrophages through the interaction of CD40L with CD40. Another group further demonstrated that monocyte-derived IL-1 and IL-6 were the key factors driving CRS in humanized SGM3 (HuSGM3) mice, and they suggested that tocilizumab did not confer protection against delayed lethal neurotoxicity. Conversely, the administration of the IL-1 receptor antagonist anakinra proved effective in mitigating both CRS and neurotoxicity, ultimately leading to a significant extension in leukemia-free survival [34]. Huang et al. observed that when CAR-T cells engaged with cancer cells, they activated Caspase-3 in the target cells and induced cancer cell pyroptosis through the activation of gasdermin E (GSDME). Consequently, this process led to the simultaneous release of factors such as ATP and HMGB1, which activated the Caspase-1-gasdermin D (GSDMD) and MAPK-NF-κB inflammatory pathways within macrophages, triggering an inflammatory factors storm [42]. The severity of CRS correlated positively with the levels of GSDME in patients-derived tumor cells. Another investigation revealed that catecholamines play a role in coordinating immune dysregulation linked to CRS through a self-amplifying loop within macrophages [43]. The deficiency of myeloid-specific tyrosine hydroxylase disrupted this loop. Elevated catecholamine levels coincided with the extensive release of cytokines, and inhibiting catecholamine synthesis reduces cytokine release both in vitro and in mice. The use of metyrosine for pharmacological catecholamine blockade protected mice from the lethal complications induced by CRS triggered by the use of CAR-T therapy [43].

### 2.2. Pathophysiological Mechanism of ICANS

ICANS, alternatively recognized as neurotoxicity, is another potential toxicity of CAR-T therapy [19]. It typically manifests within days to weeks following the infusion of CAR-T cells. Its clinical spectrum is wide-ranging, encompassing mild cognitive impairment to more severe neurological deficits. It often presents as a toxic encephalopathy, with early symptoms including confusion, diminished attention, language impairment, impaired writing function, emotional agitation, aphasia, somnolence, and tremors [35]. The American Society for Transplantation and Cellular Therapy (ASTCT) classifies adult ICANS into five grades based on the immune effector cell-associated encephalopathy (ICE) score, depressed level of consciousness, seizures, motor findings, and elevated intracranial pressure/cerebral edema [40]. It is noteworthy that ICANS is intricately linked to alterations in electroencephalographic and neuroimaging outcomes [44]. Hence, it is imperative to establish a timely and precise diagnosis of ICANS through a comprehensive assessment encompassing clinical manifestations, neuroimaging, and electroencephalography. The integration of these diagnostic modalities not only enhances the accuracy of identifying ICANS but also provides valuable insights into the underlying neural mechanisms contributing to this syndrome. Recognizing the dynamic interplay between clinical presentations, neuroimaging findings, and electroencephalographic patterns becomes pivotal in deciphering the spectrum and severity of CAR T-cell-induced neurotoxicity.

The mechanisms of ICANS are not fully elucidated at present, involving the release of inflammatory cytokines, disruption of the blood–brain barrier (BBB), infiltration of immune cells into the central nervous system, and damage to glial cells [19,35] (Figure 2). Reportedly, single-cell RNA sequencing analysis revealed the expression of CD19 in human brain barrier cells, including pericytes and vascular smooth muscle cells [45]. This raised the possibility that on-target off-tumor effects would contribute to CD19 CAR-T cell-associated ICANS. Pericytes, which mainly form the BBB, express CD19, suggesting that the mechanism of ICANS may involve CRS-induced disruption of BBB integrity [45]. This allows CAR-T cells to breach the BBB, attacking pericytes expressing CD19, leading to further breakdown of the BBB. Subsequently, a large number of CAR-T cells enter the central nervous system (CNS), causing more severe neurotoxicity.

### 2.3. Pathophysiological Mechanism of OTOT

OTOT in CAR-T therapy refers to the phenomenon where CAR-T cells inadvertently harm normal tissues and cells while targeting the intended antigen [22]. This adverse reaction occurs because the tumor-associated antigen (TAA) designed for CAR-T cells is expressed not only in tumor cells but also in normal tissues outside of the tumor, leading to attacks on normal cells by CAR-T cells [22] (Figure 3). This toxic reaction may trigger a range of adverse events, including but not limited to inflammatory reactions, tissue damage, and organ dysfunction. For example, CD19 CAR-T cells may attack normal B cells, causing B cell dysplasia [46]. In CAR-T therapy for solid tumors, the phenomenon is more pronounced due to the expression of certain target antigens to varying degrees in non-tumor tissues or organs. In a clinical study of human epidermal growth factor receptor 2 (HER2) CAR-T therapy for metastatic colon carcinoma, the patient experienced acute respiratory distress and succumbed to fatal pulmonary edema five days after cell infusion [47]. This was attributed to the CAR-T cells erroneously identifying low-level HER2 expression in the pulmonary epithelium, leading to lethal pulmonary edema.

## 3. Leveraging Biomaterials to Manage CAR-T-Related Toxicities

### 3.1. Biomaterials for the Treatment of CRS and ICANS

The clinical use of immunosuppressive glucocorticoids [48] and tocilizumab [49] may contribute to mitigating CRS. Severe ICANS is typically managed with corticosteroid therapy, while tocilizumab has shown limited efficacy in the majority of ICANS cases [50]. However, the use of tocilizumab has also raised concerns about adverse drug reactions, such as lung and liver sarcoidosis [51]. Corticosteroids, on the one hand, mediate broad anti-inflammatory and immunosuppressive effects but concurrently inhibit the proliferation of CAR-T cells and the secretion of effector cytokines. On the other hand, the extensive use of corticosteroids brings about various side effects, including the risk of osteonecrosis and infections [52,53,54]. Therefore, delineating strategies to suppress CRS and ICANS, minimizing their toxicities while preserving the in vivo proliferation and cytotoxic functions of CAR-T cells, and ensuring that the therapeutic agents employed do not induce secondary side effects have emerged as pivotal research directions in the current investigative landscape. In this regard, two preclinical studies have endeavored to address this issue through the application of biomaterials.

Excessive cytokine production occurs during the process of CRS. Therefore, the precise modulation of the immune response through the neutralization or capture of cytokines holds the potential to reduce systemic inflammation levels, alleviating the systemic inflammatory reaction caused by cytokine storms.

IL-6 plays a crucial role in CRS induced by CAR-T cell therapy, serving as a key factor contributing to inflammation and other adverse effects [55]. Li et al. proposed a temperature-sensitive hydrogel conjugated with IL-6 antibodies, termed the IL-6 sponge (IL6S), which was prepared by chemically conjugating IL-6-specific antibodies to a thermo-responsive hydrogel poly (N-isopropylacrylamide-co-methacrylic acid) [56] (Figure 4). IL6S can be administered subcutaneously prior to CAR-T cell infusion to capture the aberrantly increased IL-6 during CRS in real time. CRS models were established in SCID-beige and HuSGM3 mice. Mice were treated with PBS, IL-6 antibody, or IL6S before CAR-T cell infusion, and serum cytokines and CRS-related symptoms were continuously monitored. Compared to mice treated with PBS or IL-6 antibody, mice receiving IL6S exhibited a significant reduction in IL-6 concentration in serum. Moreover, IL6S significantly inhibited several other cytokines, including chemokine (C-C motif) ligand 2 (CCL2), TNF-α, GM-CSF, granulocyte colony-stimulating factor (G-CSF), IL-3, and IL-10. IL6S treatment significantly lowered the mortality rate of CRS mice without significantly affecting blood pressure or causing adverse reactions such as fever or weight loss [56]. Importantly, no signs of neurotoxicity (such as meningeal thickening) were observed in mice receiving IL6S compared to the PBS-treated group. Overall, these results suggest that, in a preclinical CRS mouse model, IL6S can effectively reduce the levels of inflammatory cytokines and alleviate CRS-related symptoms [56].

Ultimately, as IL6S could not degrade in vivo, this study aimed to restore mobility by covering the injection area with an ice pack, inducing a gel-to-sol transition, enabling the removal of IL6S with a syringe. Researchers have also assessed the potential long-term toxicity associated with IL6S implantation and its impact on the anti-tumor efficacy of CAR-T cells. The results demonstrated that IL6S exhibited good biocompatibility and biosafety without compromising the tumor-clearing effectiveness in a xenograft mouse model transplanted with Raji cells [56].

However, this novel strategy employed a temperature-sensitive hydrogel conjugated with antibodies for IL-6 to mitigate CRS during CAR-T therapy. While the approach shows promise, a thorough evaluation of the limitations associated with this IL-6-adsorbing hydrogel (IL6S) is crucial from a biomaterial standpoint. First of all, this study did not delve into the degradation profile of the hydrogel and the clearance of its byproducts. These aspects are critical for ensuring the safe removal of the hydrogel and minimizing any residual material or potential immunogenicity post-removal. In addition, researchers have thus far investigated the ability of IL6S to sequester IL-6 exclusively in small animal models, utilizing mice and rabbits. However, the dosage, injection sites, and optimal timing of IL6S administration may significantly differ when transitioning to large animal models and non-human primate models that more closely approximate human physiology.

The monocyte–macrophage system plays a crucial role in the development of CRS, with the hyperactive response of these cells releasing cytokines such as IL-6 and IL-1, triggering CRS and ICANS [33,34]. Therefore, a potential therapeutic approach for the treatment of CRS and ICANS could be provided by blocking the interaction between CAR-T cells and monocytes to reduce the excessive activation of monocytes/macrophages. Gong et al. ingeniously designed a strategy for the in situ PEGylation of CAR-T cells to address CRS and ICANS [57] (Figure 5). They pre-incubated CAR-T cells with Ac4ManNAz, displaying azido groups on their surfaces through bioorthogonal reaction. Subsequently, through click chemistry, they modified the T cell surface with polyethylene glycol (PEG) of different molecular weights. Further investigations revealed a significant reduction in the cytotoxicity of CAR-T cells against tumor cells when the cell surface was modified with 600K PEG. Moreover, the release of IL-6 from activated monocytes and TNF-α from activated CAR-T cells markedly decreased [57]. These findings suggest that successful PEGylation of CAR-T cells, under the spatial hindrance effect generated by high-molecular-weight PEG, effectively blocked contact with other cells, weakened monocyte activation, and reduced the release of inflammatory factors.

The researchers further explored the feasibility of in situ PEGylation of CAR-T cells in a CRS and ICANS model induced in HuSGM3 mice. They discovered that intravenously injected DBCO-PEG600K could couple with azide-modified CAR-T cells in vivo, undergoing in situ PEGylation. The experimental results indicated that doses of DBCO-PEG600K greater than 10 mg/kg could alleviate mouse CRS-associated symptoms, reverse weight loss, and lead to temperature elevation and cytokine release. Importantly, compared to tocilizumab, PEGylated CAR-T cells were effective in reducing neurotoxicity [57]. Interestingly, compared to unmodified CAR-T cells, DBCO-PEG-modified CAR-T cells collected on day 10 did not induce tumor cell killing and the release of cytokines (IL-6, IL-1, TNF-α). However, on day 15, DBCO-PEG-modified CAR-T cells induced substantial tumor cell killing and CAR-T cytotoxicity-related TNF-α release without a close association with CRS-related IL-6 and IL-1 [57]. This gradual dilution of DBCO-PEG600K on the CAR-T cell surface allowed interactions between cells to slowly recover. As CAR-T cell-tumor cell interactions recovered earlier than CAR-T cell–monocyte interactions, this approach allowed CAR-T cells to kill tumor cells before monocytes were activated, implying that this method effectively reduced side effects without compromising the clearance efficacy of CAR-T cells against tumors. Overall, this strategy ingeniously struck a balance between maximal anti-tumor efficacy and minimal adverse reactions.

The introduction of spatial hindrance through high molecular weight PEG modification via DBCO presents an innovative strategy to attenuate the interaction between CAR-T cells and monocytes, thereby alleviating CRS. Nevertheless, it is imperative to consider the immunogenicity of PEG and its impact on the immune system’s clearance of PEG. Owing to its excellent biocompatibility and tunable physicochemical properties, PEG plays a crucial role in various domains, including pharmaceuticals, biotechnology, the food industry, as well as cosmetics and personal care products. Therefore, people frequently come into contact with and use products containing PEG in their daily lives, and as a consequence, many individuals have produced antibodies against PEG [58]. Reportedly, more than 25% of healthy individuals possess anti-PEG IgM or IgG antibodies, respectively, and 8.3% concurrently harbor both anti-PEG IgM and IgG antibodies [59]. Hence, these pre-existing antibodies are likely to significantly impact the in vivo pharmacokinetics of DBCO-PEG upon infusion, as well as the efficacy of in situ PEGylation of CAR-T cells.

### 3.2. Biomaterials for Controlling CAR-T Activity and Avoiding OTOT

Currently, synthetic biology methods, such as logic-gated CAR-T cell technology, could be leveraged to regulate the activation of CAR-T cells and circumvent OTOT [22,60,61]. Nevertheless, due to the extensive overlap of antigens between tumor cells and normal tissues, it is challenging to identify ideal antigens that can distinguish between tumor and normal tissues. Additionally, pre-integrating suicide genes into the CAR structure can effectively eliminate CAR-T cells in vivo [62,63]. However, this approach faces difficulties in grasping the timing of suicide-inducing drug administration, and it may potentially impair the anti-tumor efficacy. Therefore, precise control over CAR-T activity is essential to achieve localized activation at the tumor site and reduce the toxicity of OTOT. Integrating stimulus-responsive biomaterial technology with immunotherapy to achieve remote control of CAR-T cells, enabling time and spatial-specific activation of CAR-T cells, could potentially offer a safer approach to inhibit tumor growth in the body.

Miller et al. ingeniously incorporated the HSPB1 into the CAR structure, creating a sophisticated “photothermal sensitive switch” that enables precise triggering of CAR expression [64] (Figure 6). The exciting research findings revealed that after a brief heat treatment, CAR transgene expression in primary T cells significantly increased a lot [64]. To achieve localized thermal therapy for tumors, researchers judiciously selected plasmonic AuNR as highly efficient photothermal converters, which could efficiently convert NIR (650~900 nm) into thermal energy. After intravenous injection of AuNR and CAR-T cells, it was observed that only CD19+ tumors triggered significant CAR expression after NIR irradiation. Untreated or CD19 tumors failed to activate CD19 CAR-T cells. Furthermore, the study revealed that locally induced activation of CAR-T was spatially restricted to the heated area, with no CAR-T cells observed in distant tumors and spleen, efficaciously avoiding adverse effects on non-tumor organs [64].

Another team designed a more sophisticated photoresponsive biomaterial, which was a hexagonally shaped upconversion nanoplate (UCNPs) with a core–shell composition of β-NaYbF4: 0.5%Tm@NaYF4, where Yb3+ acts as a photosensitizer receiving NIR excitation light, and Tm3+ serves as a blue light emitter [65]. Upon 980 nm NIR excitation, UCNPs could function as miniature deep-tissue photon sensors, inducing intense blue light emission. Introducing the newly engineered photoresponsive module into the CAR structure, they constructed light-switchable CAR-T cells (LiCAR-T) combined with UCNPs. Upon NIR irradiation, these cells, LiCAR-T, could induce CAR expression only under dual activation of blue light signals and tumor antigens [65]. LiCAR-T cells exhibited antigen-dependent and spatiotemporally dependent cytotoxicity against tumors. Furthermore, they established OTOT mouse models and a CRS model based on SCID-beige mice, finding that CD19 LiCAR-T cells significantly reduced B cell aplasia and decreased the cytokine level [65].

Unlike the limited penetration capability of NIR, focused ultrasound (FUS) offers a safe and non-invasive means to deliver energy into deeper tissues [66]. The method of controlling CAR-T cell activity through short, pulsed waves of FUS has been reported [67], but currently, there are no stimulus-responsive biomaterials specifically designed for this purpose. Some researchers have provided an overview anticipating the use of nanotechnology and other external physical stimuli such as magnetic fields, X-rays, electric fields, etc., to remotely manipulate the activity of CAR-T cells for the treatment of malignant tumors [68], thereby eliciting a particularly powerful therapeutic effect on solid tumors and reducing the risk of OTOT. Similarly, these external means of physical manipulation have also overcome the drawback of NIR, namely its relatively insufficient penetration depth in the context of the human body [69], particularly in tissues abundant with blood or pigments, constraining its precision in accurately localizing and monitoring deep-seated tissues.

## 4. Conclusions and Future Prospects

In summary, the landscape of CAR-T therapy is undeniably transformative, marking a paradigm shift in cancer treatment efficacy. However, notable side effects, including CRS, ICANS, and OTOT, cast a looming shadow over its widespread application. This comprehensive review has undertaken an artful expedition into the realm of biomaterials, aiming to unravel the intricate tapestry of toxicities associated with CAR-T therapy and navigate a safer course for its future.

Our exploration began with an in-depth analysis of the pathological mechanisms underlying CRS, ICANS, and OTOT. The dysregulated immune responses and the cascade of inflammatory cytokines implicated in these toxicities were scrutinized, shedding light on the complex interplay between CAR-T cells and the host immune system. Understanding the intricacies of these toxicities is paramount for the development of targeted interventions, and our discourse has provided a roadmap for this endeavor.

The central theme of our discourse has been the utilization of biomaterials as a strategic tool to mitigate the adverse effects of CAR-T therapy. From engineering hydrogels to in situ PEGylation strategies, researchers have crafted an impressive symphony of innovative approaches. These biomaterial interventions act as guardians, providing nuanced control over the activity of CAR-T cells, creating a safeguard against the unleashed storm of cytokines, and avoiding OTOT. The success of these interventions in preclinical models underscores their potential for translation into clinical settings.

As we peer into the future, the promise of biomaterial interventions in sculpting a safer and more efficacious path for CAR-T cell therapy becomes evident. The inherent flexibility of biomaterial design allows for tailored solutions, addressing specific challenges posed by CRS, ICANS, and OTOT. The integration of stimuli-responsive materials and advanced gene editing techniques further enhances the precision and safety of these interventions. In the quest to champion safety in cellular immunotherapy, the role of biomaterials extends beyond mitigating toxicities. These materials open avenues for the development of controlled-release systems, enabling precise modulation of CAR-T cell activity. Additionally, the exploration of remote control, such as focused ultrasound, adds a layer of sophistication to the therapeutic arsenal.

Most recently, the prospective application of CAR-T cell therapy in non-neoplastic diseases has garnered considerable attention [70,71,72,73]. Current research indicates that CAR-T cells, through the modulation of immune system hyperactivation, hold promise in terms of interventions intended for autoimmune disorders. This technology has demonstrated potential therapeutic efficacy in the realm of autoimmune diseases and immune-mediated pathologies, encompassing refractory systemic lupus erythematosus (SLE) [74,75,76], rheumatoid arthritis (RA) [77,78], type 1 diabetes mellitus (T1DM) [79,80,81], pemphigus vulgaris [82], muscle-specific tyrosine kinase myasthenia gravis (MuSK MG) [83], as well as colitis [84,85] and multiple sclerosis (MS) [86]. In clinical settings, CAR-T therapy has been reported in the context of diseases such as SLE [75], anti-synthetase syndrome [87], and systemic sclerosis (SSc) [88]. Remarkably, treatment-related adverse effects, notably CRS, have manifested in patients undergoing these therapeutic interventions. These findings have generated widespread interest in the current academic milieu, offering valuable insights for the expanded application of CAR-T cell therapy in the treatment of diverse autoimmune conditions. Nonetheless, it is essential to underscore that further in-depth research and clinical practices are requisite to ensure the safety and efficacy of this technology. The documented cases underscore the imperative for ongoing scrutiny and refinement of CAR-T therapies in the clinical management of these conditions. As the therapeutic landscape evolves, meticulous monitoring and further investigations are warranted to enhance the safety and efficacy profile of CAR-T treatments in the realm of autoimmune disorders. Therefore, concurrently alleviating the toxicity associated with anti-tumor CAR-T treatments leveraging biomaterials while also extending the application of these to overcome adverse reactions in the treatment of autoimmune and immune-mediated pathologies and exploring the feasibility thereof represent a crucial avenue for future endeavors in advanced medical research.

Looking to the future, to begin with, from a biomaterial perspective, future efforts should primarily focus on conducting comprehensive safety evaluations of biomaterials in large animal models and non-human primates, which exhibit physiological characteristics more closely resembling those of humans. This includes an in-depth assessment of safety profiles in disease models. Simultaneously, investigations into appropriate administration routes, dosage, and timing in these models are essential. Additionally, careful consideration must be given to the immunogenicity of biomaterials and the impact of immune responses on their in vivo efficacy. These aspects require thorough evaluation to pave the way for successful clinical translation. Additionally, the translation of these innovative strategies from bench to bedside holds immense promise. Clinical trials investigating the safety and efficacy of biomaterial interventions in CAR-T therapy are on the horizon. Collaborative efforts between biomaterial scientists, immunologists, and clinicians will be pivotal in shaping the future of CAR-T therapy, ensuring its broader accessibility and minimizing the risks of adverse events.

In conclusion, the judicious integration of biomaterials into the realm of CAR-T therapy paints an optimistic outlook for the future. The artful expedition we have undertaken signifies not only a profound understanding of the challenges posed by toxicities but also a commitment to overcoming these challenges with ingenuity and precision. As we navigate this uncharted territory, the synergy between advanced medical science and biomaterial innovations emerges as a beacon of hope, paving the way for a safer and more effective era in cellular immunotherapy. The journey continues, and the promise of a transformative impact on cancer treatment remains within reach.

## Data Availability

The data presented in this study are available on request from the corresponding author.

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
