# Peer review of "Harnessing Biomaterials for Safeguarding Chimeric Antigen Receptor T Cell Therapy: An Artful Expedition in Mitigating Adverse Effects"

_pharmaceuticals, 2024, doi:10.3390/ph17010139_

Round 1
Reviewer 1 Report
Comments and Suggestions for Authors
The authors aim to describe the common side effects of CAR T cell therapies (CRS, OTOT, and ICANS), to describe the pathophysiological mechanisms, and to present conventional (CS, IL6) and novel therapeutic strategies to overcome these side effects.
This review critically examines the anti-tumor effect of thermoresponsive hydrogels (IL6 sponge), CAR T-azide and CAR T-PEG interaction, and the subsequent time-limited PEG dilution on CAR T cells, which also prevents macrophage activation, about CRS and ICANS.
The anti-OTOT effects of logic-gated, photothermal CAR-T, LiCART, and FUS technologies are also illustrated.
The topic and structure of the article are excellent; the techniques presented are very novel and have great future potential.
I would suggest one small addition from future perspectives: the technologies used in antitumor CAR T treatments seem promising. Could they also alleviate side effects in autoimmune and immune-mediated pathologies?
Reviewer 2 Report
Comments and Suggestions for Authors
The review titled "Harnessing Biomaterials for Safeguarding CAR-T Therapy: An Artful
Expedition in Mitigating Adverse Effects" is interesting but is not extensive. The article covers the toxicities arising out of CART therapy and how the toxicities can be mitigated by biomaterials. The figures and the topic of review is interesting and well-thought. The review covers the toxicities and the biomaterials in an overview and does not dwell into the details. I suggest that authors expand the review with more information on the toxicities and also consider the changes suggested below.
1. There are previous reviews on the toxicities of CART therapy ( PMC7785104; 10.1097/HS9.0000000000000186 ) and several more. How is this review different? The other reviews are extensive and covers more details. The authors should either expand the review with more details.
2. Page 4: "critical constituented.." Not sure what this means.
3. The authors should expand all the acronyms and abbreviations at the first instance. this is missing in several places.
4. Page 6: "Advanced biomaterials designed to adsorb virus....." this line is unnecessary and does not fit into the flow of the review. The line should be removed in my opinion.
5. The limitations of the biomaterials and how this can be improved is missing in all subsections of section 3. The authors should critically analyse this portion and report the limitations. The ways to overcome the limitations should be discussed in section 5.
6. There is not section 4 between 3 and 5. Correct the section numbering accordingly.
7. Page 10: "CD19-tumors" please insert a space to make it clear that they are CD19 negative and not a hyphen.
8. Scheme 5: The lines connecting CART peg and tumor cells have a red cross on them which shows that they do not act on tumor cells. It is my understanding that the tumor cells are recognized (though late) and eliminated specifically by pegy-lation. The presence of red crosses create confusion therefore. Please correct the same or be more specific in the figure caption.
Comments on the Quality of English Language
The article required minor English editing.
Round 2
Reviewer 2 Report
Comments and Suggestions for Authors
The authors have addressed the comments satisfactorily